# Pilot randomized trial of the effect of antibacterial mouthwash on muscle contractile function in healthy young adults

Edgar J. Gallardo[1,☉], William S. Zoughaib[1,☉], Ahaan Singhal[2], Richard L. Hoffman[1], Andrew R. Coggan[1,3]*

1 Department of Kinesiology, School of Health & Human Sciences, Indiana University Indianapolis, Indianapolis, Indiana, United States of America, 2 Department of Medicine, Indiana University School of Medicine, Indianapolis, Indiana, United States of America, 3 Indiana Center for Musculoskeletal Health, Indiana University School of Medicine, Indianapolis, Indiana, United States of America

☉ These authors contributed equally to this work.
* acoggan@iu.edu

## Abstract

Antiseptic mouthwash use is widespread due to its oral health benefits. However, its impact on systemic physiological processes, particularly nitric oxide (NO) bioavailability and muscle contractility, is not fully understood. We sought to determine the effects of cetylpyridinium (antibacterial) versus sodium chloride (control) mouthwashes on salivary and breath NO markers and muscle contractile function in healthy young adults. Thirty participants (n = 15/group) completed a randomized, parallel-arm, blinded trial, comparing the two mouthwashes before and after 7 d of treatment. NO bioavailability was inferred via measurement of salivary nitrate ($NO_3^-$), nitrite ($NO_2^-$), and cyclic guanyl monophosphate (cGMP) concentrations and breath NO level. Contractile function of the knee extensor muscles was determined via isokinetic dynamometry. No changes in salivary $NO_3^-$, $NO_2^-$, or cGMP or in breath NO were observed in response to either treatment. However, cetylpyridinium mouthwash reduced the percentage of $NO_2^-$ in saliva (17±10% vs. 25±13%; p = 0.0036). Peak torque at velocities of 0–6.28 rad/s was unaffected by mouthwash use. Calculated maximal knee extensor velocity (Vmax) and power (Pmax) were therefore also unchanged. Cetylpyridinium mouthwash reduces the relative abundance of $NO_2^-$ in the oral cavity but does not significantly diminish overall NO bioavailability or impair muscle contractile function in healthy young adults.

## Introduction

Approximately $7B worth of antiseptic/antibacterial mouthwash is sold globally each year [1], with roughly ⅓ to ½ of adults in many countries using such products on a regular basis [2–4]. Mouthwash use can diminish plaque formation and reduce gingivitis [5], but such products may also have detrimental effects. Specifically, strong antibacterial mouthwash can disrupt the nitrate ($NO_3^-$) -reducing bacteria normally found in the mouth [6–9]. These symbionts reduce salivary $NO_3^-$ to nitrite ($NO_2^-$), which after being absorbed can be further reduced to form nitric oxide (NO) [10,11]. NO is a key signaling molecule involved in a variety of physiological responses,

**Data availability statement:** All relevant data are within the manuscript and its Supporting Information files.

**Funding:** This study was supported by the Diversity Scholars Research Program of the Center for Research and Learning at Indiana University Indianapolis awarded to EJG. The funders had no role in study design, data collection and analysis, decision to publish, or preparation of the manuscript.

**Competing interests:** The authors have declared that no competing interests exist.

including, but not limited to, the regulation of blood flow/pressure. As a result, interruption of this enterosalivary pathway of NO production via twice-daily use of chlorhexidine-containing mouthwash for 3–7 d has been reported to decrease plasma $NO_2^-$ levels by 15–25% and increase systolic blood pressure by 2–3 mmHg [12,13]. The latter change may be clinically significant, as it could increase risk of death from ischemic heart disease and stroke by 7% and 10%, respectively [14]. In fact, frequent use of mouthwash has been associated with the development of hypertension [15] and pre-diabetes/diabetes [16]. Chlorhexidine-containing mouthwash has also been shown to block post-exercise reductions in blood pressure [17]. Chronic mouthwash use therefore may have unintended negative consequences.

In addition to influencing blood flow/pressure, NO is involved in the regulation of numerous other physiological functions, including skeletal muscle contractile function [18]. In this regard, numerous studies in recent years have demonstrated that acute ingestion of $NO_3^-$-rich beetroot juice can enhance muscle contractility in a wide variety of individuals [cf. Ref. [19] for review]. Although the exact mechanism responsible is still unclear, this improvement has been hypothesized to result from NO-mediated changes in calcium [$Ca^{2+}$] release and/or sensitivity within muscle [20]. Indeed, muscle function seem to be more sensitive to alterations in NO availability than blood pressure, as dietary $NO_3^-$ supplementation has been routinely observed to enhance the former even in the absence of changes in the latter [21–24]. It is not known, however, whether a *reduction* in NO bioavailability due to use of antibacterial mouthwash can *inhibit* muscle function. Such knowledge is potentially relevant because muscle strength, speed, and especially power are important determinants of both athletic performance [25] and the ability to perform normal activities of daily living [26].

The purpose of the present parallel-arm study was therefore to determine the effects of antibacterial mouthwash on muscle contractile function in healthy young adults.

## Methods

### Participants

Potential participants were recruited via word-of-mouth and flyers placed around the university campus. Exclusion criteria were age < 18 or > 30 y; current use of mouthwash, antibiotics, or tobacco products; resting blood pressure > 140/90; an answer of yes to any of the questions of the Physical Activity Readiness Questionnaire (PAR-Q); or inability to provide informed consent. All other persons were included in the study. Data collection took place in two phases, with n = 6 participants studied between November 1st 2019 and March 13th 2020 and n = 24 between November 5th 2020 and March 20th 2023. All participants studied during the second phase were required to submit to weekly COVID-19 mitigation testing and then later to be vaccinated as a condition of continued university enrollment. Each participant provided written, informed consent, and the study protocol was approved by the Human Subjects Office at Indiana University (protocol number 1907204336). The study was registered at ClinicalTrials.gov (NCT04095442). Participant flow through the study is shown in Fig 1.

### Experimental design and protocol

After the initial visit to the Exercise Physiology Laboratory at Indiana University Indianapolis, which included completion of the Physical Activity Readiness Questionnaire (PAR-Q) and practicing the isokinetic dynamometer testing protocol described below, participants were randomized in blocks of four into one of two groups. The randomization scheme was generated using randomization.com by an individual (RLH) not involved in participant recruitment, testing, or data analysis, with the key concealed until completion of the study. One group was studied immediately before and after using Cepacol® mouthwash (Reckitt

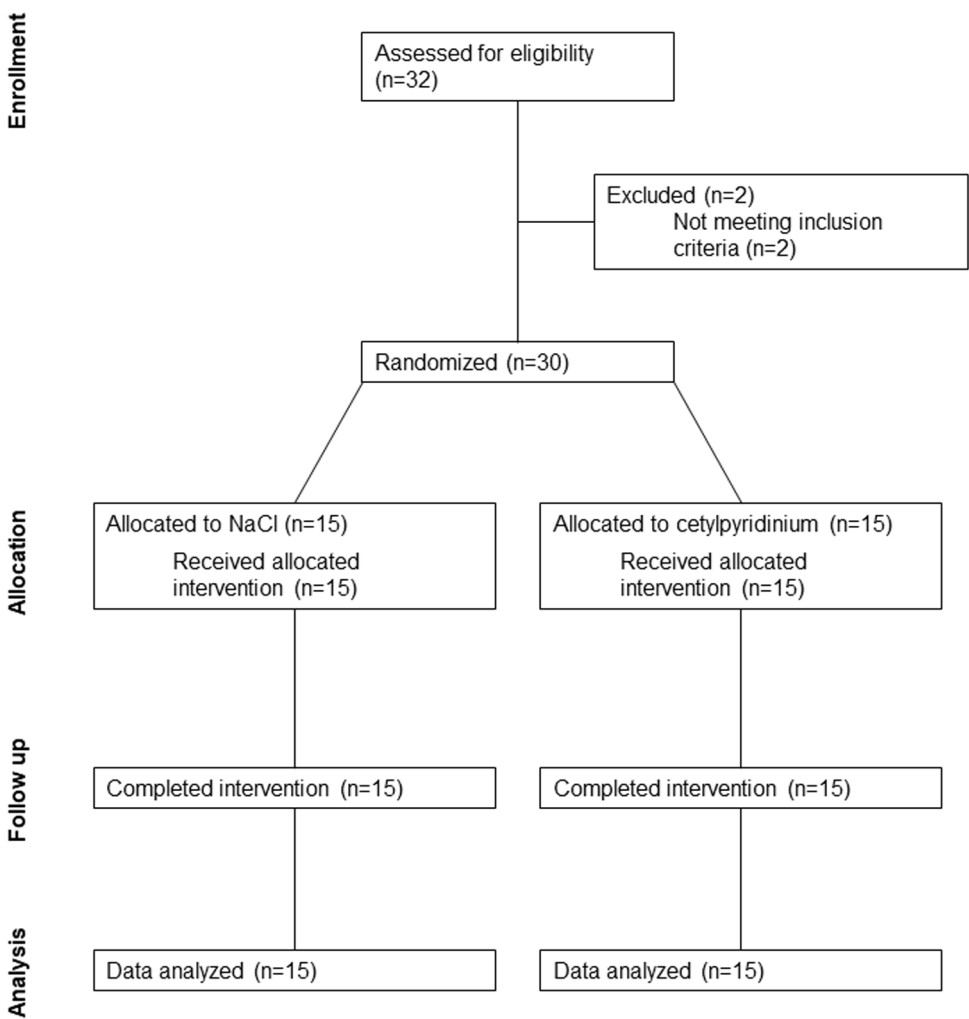

**Fig 1. CONSORT diagram.**

Benckiser, Parsippany, NJ) for 7 d. This product was chosen as the strongest non-prescription antibacterial mouthwash available in the United States [27]. As a comparison/control, a second group was studied immediately before and after using Tom's of Maine® Sea Salt Natural Mouthwash (Kennebunk, ME) for 7 d. This product was not expected to have any significant effects on the oral microbiota and hence NO production [28]. These two groups will henceforth be referred to by the active ingredients of the two products, i.e., cetylpyridinium and sodium chloride (NaCl), respectively. The physical characteristics of the participants in each group are shown in Table 1. Twenty-six (87%) of the participants were non-Hispanic White, three (10%) were Black or African American, and one (3%) was Hispanic or Latino, reflective of the racial and ethnic makeup of the local population.

Participants in both groups were instructed to refrain from exercising and to avoid alcohol, caffeine, chewing gum, and foods high in $NO_3^-$ (e.g., spinach, beets) for 24 h prior to study visits. Upon reporting to the laboratory, the participant's breath NO level, a sensitive indicator of changes in whole-body NO production [29], was first measured using a portable electrochemical analyzer (NIOX VERO®, Circassia Pharmaceuticals, Mooresville, NC). A stimulated

**Table 1. Participant characteristics.**

|  | Group | |
|---|---|---|
|  | **NaCl** | **Cetylpyridinium** |
| Total N | 15 | 15 |
| Non-Hispanic White/Black or African American/Hispanic or Latino | 14/1/0 | 12/2/1 |
| Sex (M/F) | 8/7 | 9/6 |
| Age (y) | 22 ± 3 | 23 ± 4 |
| Height (m) | 1.74 ± 0.13 | 1.71 ± 0.05 |
| Mass (kg) | 80.1 ± 15.2 | 76.6 ± 11.3 |
| BMI (kg/m²) | 26.5 ± 3.7 | 26.1 ± 3.3 |

Values are means ± SD.

saliva sample was then collected using a commercial collection kit (Salivette, Sarstedt, Newton, NC). (Stimulated saliva was collected because it is more reflective of plasma $NO_3^-$ and $NO_2^-$ levels than unstimulated saliva [30].) Due to pandemic-related research restrictions, we were unable to obtain blood samples.) Participants briefly chewed on a synthetic swab until it was saturated with saliva. The swab was then immediately centrifuged for 2 min at 1000 g and 4° C and the saliva frozen at −20 °C until subsequently analyzed for $NO_3^-$ and $NO_2^-$ concentrations via high performance liquid chromatography (ENO-30, Amuza, San Diego, CA). Salivary cCGMP levels (which are highly correlated with plasma concentrations [31]) were also determined in these samples using a commercial enzyme-linked immunosorbent assay kit (Item number 581021, Cayman Chemical, Ann Arbor, MI). The coefficient of variation for these triplicate measurements averaged 12.0 ± 9.8%.

The contractile properties of the quadriceps muscle group were next determined using an isokinetic dynamometer (Biodex System 4 Pro, Biodex Medical Systems, Shirley, NY) as previously described [21–24]. Briefly, participants performed three maximal knee extensions at angular velocities of 0, 1.57, 3.14, 4.71, and 6.28 rad/s, with 2 min of rest allowed between each set of contractions. The highest torque generated at each velocity was used to calculate peak power at that velocity, after which the maximal velocity (Vmax) and power (Pmax) of knee extension were determined by fitting a parabolic function to the data.

Upon completion of the baseline visit, participants were provided with a bottle of their assigned mouthwash in an opaque bag (to conceal the assignment from the investigators). They were instructed to rinse their mouth as directed on the product's packaging for 30 s twice per day (i.e., morning and evening, including on the morning of study) for 7 d and to record usage of the mouthwash on a provided form. They were also instructed to maintain their normal oral hygiene habits (e.g., toothbrushing, flossing) but to not use any other mouthwash products during this period. Note that although the two products differed significantly in terms of packaging, color, taste, etc., the participants were kept unaware of the other (i.e., non-assigned) product and were simply told that the purpose of the study was to compare two different mouthwashes, with no expectation as to the outcome. After 7 d, they returned to the laboratory whereupon the above measurements were repeated.

## Statistical analyses

Statistical analyses were performed using GraphPad Prism version 10.2.3 (GraphPad Software, La Jolla, CA). Normality of data distribution was tested using the D'Agostino-Pearson omnibus test, whereas Grubb's test was used to check for possible outliers. Salivary $NO_3^-$, $NO_2^-$, and log-normalized cGMP concentrations, breath NO levels, and Vmax and Pmax were

analyzed using two-way repeated measures analysis of variance (ANOVA), with treatment as a between-subject factor and time as a within-subject factor. Isometric and isokinetic knee extensor peak torque data were analyzed using a three-way repeated measures ANOVA, with treatment as a between-subject factor while velocity and time were treated as within-subject factors. Post-hoc testing was performed using the Holm-Šidák multiple comparison procedure. A two-tailed $P < 0.05$ was considered significant. Pmax was defined as the primary outcome; all other outcomes were secondary. As this study was the first of its kind, we lacked the preliminary data necessary to perform a formal sample size/power analysis. However, our chosen sample size should have allowed us to detect any change in Pmax with an effect size (i.e., Cohen's f [32]) ≥ 0.15 with an alpha of 0.05 and 1-beta of 0.95. According to Cohen [32], this would be between a "small" (f = 0.10) and a "medium" (f = 0.25) effect size.

## Results and discussion

All eligible participants completed the intervention, with no adverse events in either the NaCl or cetylpyridinium groups. In particular, none experienced a COVID-19 infection during their participation, none reported that they were suffering from long COVID, and none reported any reason why they should not perform physical activity (PAR-Q question 7). However, saliva was available for analysis in only n = 13/group (n = 11 for cGMP in the cetylpyridinium group), whereas due to equipment obsolescence breath NO was only measured in n = 8/group. No other data were missing.

### Markers of NO bioavailability

No significant changes were observed in salivary $NO_3^-$ or $NO_2^-$ concentrations or in their sum in either group (Figs 2A-C). There were also no significant changes in salivary cGMP concentrations or in breath NO levels (Figs 2E, 2F). There was, however, a significant reduction in the relative abundance of $NO_2^-$ in the cetylpyridinium group, which decreased from $25 \pm 13\%$ to $17 \pm 10\%$ of the total salivary $NO_3^-$ plus $NO_2^-$ concentration (t = 3.512, DF = 24, multiplicity adjusted P = 0.0036) (Fig 2D). The largest decline (i.e., from 54% to 20%) was observed in the individual with the highest initial value. Although not a statistical outlier, the data were nonetheless also analyzed excluding these results. The reduction in salivary $NO_2^-$ in the other 12 participants (i.e., from $22 \pm 11\%$ to $17 \pm 11\%$) was still highly significant (t = 3.245, DF = 23, multiplicity adjusted P = 0.0071).

### Muscle contractile function

Although there was a highly significant effect of velocity on knee extensor peak torque, the effects of time and treatment were not significant, nor were there any significant interaction effects (Table 2). There were therefore no significant changes in either Vmax (Fig 3A) (time: $F_{1,28} = 0.0007483$, P = 0.9784; treatment: $F_{1,28} = 2.059$, P = 0.1624; time x treatment: $F_{1,28} = 0.4950$, P = 0.4875) or Pmax (Fig 3B) (time: $F_{1,28} = 0.03467$, P = 0.8536; treatment: $F_{1,28} = 0.6123$, P = 0.4405; time x treatment: $F_{1,28} = 0.9878$, P = 0.3288).

The purpose of this investigation was to determine whether use of over-the-counter antibacterial mouthwash would reduce NO production via the enterosalivary pathway and thus impair muscle contractility in healthy young adults. Contrary to this hypothesis, we found no significant changes in salivary $NO_3^-$, $NO_2^-$, or cGMP concentrations, in breath NO levels, or in the maximal voluntary isometric/isokinetic torque or speed or power of knee extension. This is the first study to specifically examine the effects of mouthwash use on muscle contractile function or, to our knowledge, any determinant of human exercise performance.

There are several possible explanations for these negative findings. The first is simply that the participants failed to utilize the provided mouthwash as instructed. However, review of

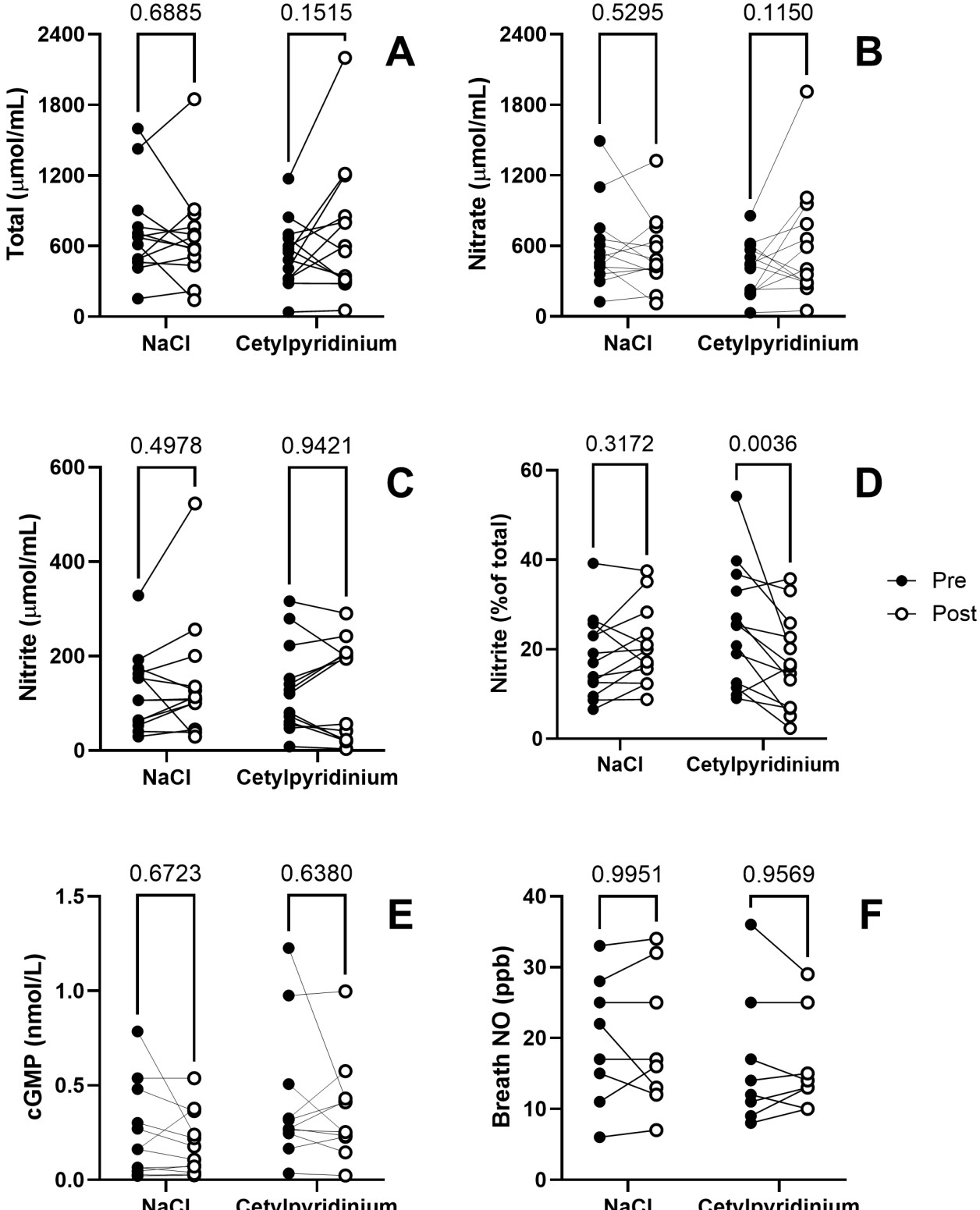

**Fig 2. Effect of NaCl- or cetylpyridinium-containing mouthwash on markers of NO bioavailability.**

**Table 2. Effect of NaCl- or cetylpyridinium-containing mouthwash on the maximal isometric/isokinetic torque of the knee extensors.**

| | Group | | | | |
| --- | --- | --- | --- | --- | --- |
| | NaCl | | Cetylpyridinium | | |
| Velocity[rad/s] | Pre | Post | Pre | Post | ANOVA results |
| 0 | $2.65 \pm 0.53$ | $2.74 \pm 0.55$ | $2.51 \pm 0.81$ | $2.46 \pm 0.82$ | Velocity: $F_{1.607,45} = 125.5$, $P < 0.0001$ |
| 1.57 | $2.05 \pm 0.51$ | $2.14 \pm 0.51$ | $2.09 \pm 0.66$ | $2.05 \pm 0.63$ | Time: $F_{1,28} = 0.2189$; $P = 0.6247$ |
| 3.14 | $1.68 \pm 0.46$ | $1.70 \pm 0.41$ | $1.70 \pm 0.59$ | $1.61 \pm 0.57$ | Treatment: $F_{1,28} = 0.1416$; $P = 0.7095$ |
| 4.71 | $1.35 \pm 0.40$ | $1.37 \pm 0.46$ | $1.34 \pm 0.48$ | $1.33 \pm 0.47$ | Velocity x Group: $F_{4,122} = 0.5770$, $P = 0.6799$ |
| 6.28 | $1.07 \pm 0.39$ | $1.13 \pm 0.45$ | $1.06 \pm 0.45$ | $1.10 \pm 0.40$ | Velocity x Time: $F_{1.731,48.46} = 0.661$, $P = 0.4975$ |
| | | | | | Group x Time: $F_{1,28} = 2.785$, $P = 0.1063$ |
| | | | | | Velocity x Group x Time: $F_{4,112} = 0.5019$, $P = 0.7344$ |

Values are means ± SD in Nm/kg.

the participants' logs indicated excellent compliance, averaging $97 \pm 7\%$ and $97 \pm 4\%$ in the cetylpyridinium and NaCl groups, respectively. Furthermore, the relative abundance of $NO_2^-$ in saliva was significantly reduced in the cetylpyridinium group, as discussed in greater detail below. Thus, while participant non-compliance cannot be completely excluded as a factor, this explanation seems unlikely.

The second possibility is that twice-daily use of an over-the-counter antibacterial mouthwash for 7 d failed to alter the oral microbiome sufficiently to have any "downstream" consequences. This could be because the bacteria found in the oral cavity are extremely resilient, rapidly repopulating themselves after disruption via physical (i.e., toothbrushing) or chemical (i.e., mouthwash) means [33]. Nonetheless, cetylpyridinium-containing mouthwashes have been shown to suppress salivary bacterial counts by 40–70% for up to 6 h [27] and are highly effective at suppressing growth of and/or eradicating bacteria from biofilms *ex vivo* [27,33–36]. Cetylpyridinium-based mouthwash has also been found to markedly blunt increases in salivary and plasma $NO_2^-$ following an oral $NO_3^-$ load [37]. Consistent with these prior observations, in the present study there was a significant (t = 3.512, DF = 24, multiplicity adjusted P = 0.0036) reduction in the percentage of $NO_2^-$ in saliva after 7 d of treatment in the cetylpyridinium group. However, there were no changes in the absolute concentrations of either $NO_2^-$ or $NO_3^-$, suggesting that the effects of cetylpyrdinium on the oral microbiome were somewhat limited.

The third possibility is that cetylpyridinium-containing mouthwash did in fact markedly inhibit NO production via the enterosalivary pathway, but any reduced contribution from this source was simply too small to matter. Consider the following: in the US, normal dietary $NO_3^-$ intake is 0.5–1.5 mmol/d [38,39]. However, it has been estimated that only approximately 5% of ingested $NO_3^-$ is converted to $NO_2^-$ [38–41]. Even assuming that all of this newly-formed $NO_2^-$ is ultimately reduced to NO, this implies a maximum rate of NO formation from ingested $NO_3^-$ of 0.5–1.5 x 0.05 = 0.025–0.075 mmol/d. At the same time, however, in healthy individuals NOS-mediated NO synthesis is approximately 1 mmol/d [42]. It is therefore clear that the enterosalivary pathway is normally a relatively minor source of NO, a conclusion supported by previous isotopic tracer research [43,44]. As such, use of antibacterial mouthwash might not be expected to have any marked biological effects unless combined with/compared to ingestion of large amounts of $NO_3^-$ [37,45,46] (see below).

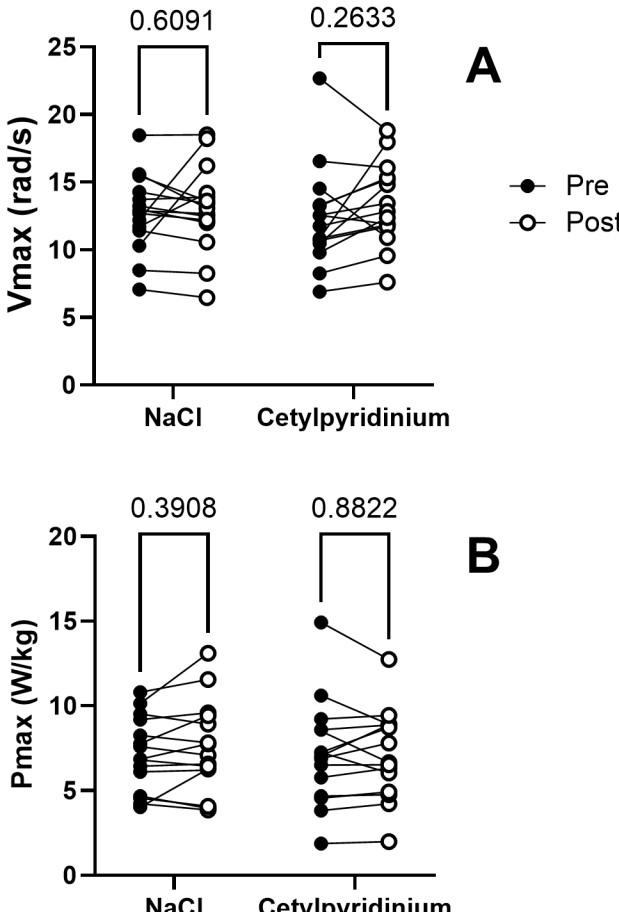

**Fig 3. Effect of NaCl- or cetylpyridinium-containing mouthwash on maximal knee extensor velocity (Vmax) and power (Pmax).**

As stated previously, to our knowledge this is the first study to determine the influence of mouthwash use *per se* on muscle contractility or any other aspect of physical function/exercise capacity. A handful of previous investigations, however, have examined the effects of chlorhexidine-containing mouthwash (which is prescription-only in the US, Canada, and some other countries), sometimes in conjunction with a $NO_3^-$-restricted diet, on other physiological parameters, particularly blood pressure [8,9,12,13]. Compared to the present results, these studies have generally found much larger changes in salivary $NO_2^-$ and $NO_3^-$ concentrations, averaging −69% and +80%, respectively, presumably reflecting the more prolonged antibacterial action of chlorhexidine vs. cetylpyridinium [27,33,36]. However, salivary and plasma concentrations of $NO_2^-$ are only weakly correlated [8], and changes in plasma $NO_2^-$ concentration in response to chlorhexidine use have been much more modest (i.e., −19% on average) [8,9,12,13,47]. Furthermore, no changes have been found in plasma cGMP levels [9,13], a highly sensitive indicator of whole-body NO bioavailability [29,48]. Blood pressure has also been mostly unchanged [8,9,47], with only Kapil et al. [12] and Bondonno et al. [13] reporting small, but statistically significant, increases in systolic blood pressure and only Kapil

et al. [13] finding a statistically significant increase in diastolic blood pressure. Again, this could be due to the quantitatively minor contribution of the enterosalivary pathway to NO production under normal circumstances, i.e., at typical levels of dietary $NO_3^-$ intake and in the absence of chronic diseases such as hypertension or heart failure, in which NOS-mediated NO synthesis is reduced [cf. Ref. [42] for review].

There are limitations to the present study. As implied above, we might have obtained different results had we utilized a mouthwash containing chlorhexidine instead of cetylpyridinium, had the participants rinse their mouths more frequently than the recommended two times per day, or had tested the effects of mouthwash in conjunction with a low $NO_3^-$ diet or following $NO_3^-$ ingestion. Our results might also have been different had we examined some other exercise outcome, e.g., endurance performance ability, instead of muscle contractile function, and/or studied a population in whom NO bioavailability is already reduced (e.g., older individuals). Although salivary [49] (and plasma [50] and urinary [51]) $NO_3^-$ and $NO_2^-$ levels do not exhibit any diurnal variation, muscle power does [52], such that our findings might still have been different if we had tested at different times of day. We also did not determine the aerobic fitness or body composition of our participants, and cannot rule out an impact of long COVID on their responses. Finally, we only measured the concentrations of $NO_3^-$, $NO_2^-$, and cGMP in saliva, such that the impact of the intervention on the concentrations of these metabolites in plasma (or more importantly, muscle) is unknown. Inclusion of such measurements, however, would not have changed the fact that our primary outcome, i.e., muscle function, was unaltered. Likewise, quantification of the effects of antibacterial mouthwash on oral $NO_3^-$ reducing capacity [53] – which, notably, are *not* predictive of changes in plasma $NO_2^-$ levels [54] - would not have altered our other measured outcomes or their interpretation.

## Conclusions

This is the first study to specifically determine the effects of antibacterial mouthwash on muscle contractile function. Although mouthwash use significantly reduced the relative abundance of $NO_2^-$ in saliva, consistent with inhibition of the enterosalivary pathway, there were no changes in the concentrations of $NO_3^-$, $NO_2^-$, and cGMP in saliva and the level of NO in breath, or in maximal muscular strength, speed, or power. Short-term use of cetylpyridinium-based mouthwash does not seem to impair whole-body NO bioavailability or muscle contractility in healthy young individuals.

## Supporting information

**S1 Data. Statistical analyses.**
(DOCX)

**S2 Data. Mouthwash study data (final) .**
(XLSX)

## Author contributions

**Conceptualization:** Andrew R. Coggan.

**Data curation:** Edgar J. Gallardo, William S. Zoughaib.

**Formal analysis:** Andrew R. Coggan.

**Funding acquisition:** Edgar J. Gallardo.

**Investigation:** Edgar J. Gallardo, William S. Zoughaib, Ahaan Singhal, Richard L. Hoffman, Andrew R. Coggan.

**Project administration:** Richard L. Hoffman, Andrew R. Coggan.

**Supervision:** Andrew R. Coggan.

**Writing – original draft:** Edgar J. Gallardo, William S. Zoughaib, Ahaan Singhal, Richard L. Hoffman, Andrew R. Coggan.

**Writing – review & editing:** Edgar J. Gallardo, William S. Zoughaib, Ahaan Singhal, Richard L. Hoffman, Andrew R. Coggan.

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
