## [Decision Letter · Decision Letter 0]

28 Aug 2024

PONE-D-24-32002Randomized trial of the effect of antibacterial mouthwash on muscle contractile function in healthy young adultsPLOS ONE

Dear Dr. Coggan,

Thank you for submitting your manuscript to PLOS ONE. After careful consideration, we feel that it has merit but does not fully meet PLOS ONE’s publication criteria as it currently stands. Therefore, we invite you to submit a revised version of the manuscript that addresses the points raised during the review process.

We look forward to receiving your revised manuscript.

Kind regards,

Ajaya Bhattarai

Academic Editor

PLOS ONE

Journal Requirements:

1. When submitting your revision, we need you to address these additional requirements. Please ensure that your manuscript meets PLOS ONE's style requirements, including those for file naming. The PLOS ONE style templates can be found at https://journals.plos.org/plosone/s/file?id=wjVg/PLOSOne_formatting_sample_main_body.pdf and https://journals.plos.org/plosone/s/file?id=ba62/PLOSOne_formatting_sample_title_authors_affiliations.pdf 2. We note that the grant information you provided in the ‘Funding Information’ and ‘Financial Disclosure’ sections do not match.  When you resubmit, please ensure that you provide the correct grant numbers for the awards you received for your study in the ‘Funding Information’ section. 3.Thank you for stating the following financial disclosure: "EJG was supported by the Diversity Scholars Research Program of the Center for Research and Learning at Indiana University Indianapolis." Please state what role the funders took in the study.  If the funders had no role, please state: "The funders had no role in study design, data collection and analysis, decision to publish, or preparation of the manuscript." If this statement is not correct you must amend it as needed. Please include this amended Role of Funder statement in your cover letter; we will change the online submission form on your behalf. 4. Thank you for stating the following in the Acknowledgments Section of your manuscript: "EJG was supported by the Diversity Scholars Research Program of the Center for Research and Learning at Indiana University Indianapolis." We note that you have provided funding information that is not currently declared in your Funding Statement. However, funding information should not appear in the Acknowledgments section or other areas of your manuscript. We will only publish funding information present in the Funding Statement section of the online submission form. Please remove any funding-related text from the manuscript and let us know how you would like to update your Funding Statement. Currently, your Funding Statement reads as follows: "EJG was supported by the Diversity Scholars Research Program of the Center for Research and Learning at Indiana University Indianapolis." Please include your amended statements within your cover letter; we will change the online submission form on your behalf. 5. Please include a separate caption for each figure in your manuscript. 6. Please include captions for your Supporting Information files at the end of your manuscript, and update any in-text citations to match accordingly. Please see our Supporting Information guidelines for more information: http://journals.plos.org/plosone/s/supporting-information.

Additional Editor Comments:

The minor revision is needed.

Reviewers' comments:

Reviewer's Responses to Questions

**Comments to the Author**

1. Is the manuscript technically sound, and do the data support the conclusions?

Reviewer #1: Yes

Reviewer #2: Partly

2. Has the statistical analysis been performed appropriately and rigorously? 

Reviewer #1: Yes

Reviewer #2: Yes

3. Have the authors made all data underlying the findings in their manuscript fully available?

Reviewer #1: Yes

Reviewer #2: Yes

4. Is the manuscript presented in an intelligible fashion and written in standard English?

Reviewer #1: Yes

Reviewer #2: Yes

5. Review Comments to the Author

Reviewer #1: General Comments: The authors have provided a well written manuscript focused on investigating the effect of mouthwash on oral biomarkers of nitric oxide bioavailability and lower extremity skeletal muscle function. Although the question is of relevance given the increasing interest in nitrate supplementation, the population for testing their hypothesis is questionable. Given the support provided in the introduction and the exclusionary criteria of several of the author's manuscripts, it would seem that young healthy adults would not benefit from nitrate supplementation, possibly due to the robustness of physiological reserves at that developmental stage. Thus, the blunting of the enterosalivary pathway with mouthwash would need to have large effect to even expect a reduction in skeletal muscle function in this population. As the authors allude in the discussion, studying this effect in older adults or patient populations that have documented poor NO bioavailability would seem to be the ideal population, especially given emerging evidence of links between oral health and frailty. Nonetheless, in accordance with PLOS ONE publication criteria, the study was conducted well, and I only have minor edits about the framing of the results, questionable statistical analyses, and the preliminary nature of the study.

Specific:

Title Page: This should be labelled a pilot randomized trial for several reasons: 1) No sample size calculation was performed and the effect size from the meta-analysis includes older adults, who we would expect to have a large effect; 2) only 13/15 provided saliva, no blood NO biomarkers, and only 8/15 provided breath NO; 3) The one significant result is potentially driven by an outlier (well discuss further in methods).

Abstract: The authors state "Cetylpyridinium mouthwash INHIBITS reduction of nitrate to nitrite...". This was not tested in this study. The mouthwash was associated with a reduction in salivary NO precursors. Please edit accordingly.

- Keywords should not duplicate words used in the title. Please revise.

Methods:

- Inclusion of participants during the COVID 19 pandemic would necessitate record of if they had previously been exposed and potientially controlling for any differences. Long covid impacts are still being tested but some studies suggest initial NO altering effects [PMID: 37657532]. Please add this as a limitation of your study.

Table 1: Please add Race and Age data. Minorites reports higher exposure to societal stressors that could impact endogenous NO bioavailability. Also, baseline fitness level and body composition were not considered. Given the limitation of BMI and small sample size, differences in aerobic capacity, lean mass, and adipose could impact NO responses. If data is not available, please add as a limitation.

Statistical analyses - a sensitivity analysis should be conducted on data provided in figure D. The reader should know if removal of that one individual in the Cetylpyridinium group with the highest pre value changes the statistical significance of the result. Secondly, the effect size cited should be labeled (i.e., is it Cohen's d, partial eta, etc.) and the reader should know why it was selected.

Results: The focus on percent change is a bit misleading when the individual data points seem to show that one data point is driving the huge change. Please accompany this with the sensitivity analysis data.

Discussion: Given the limitations the findings should be toned down. Please remove any mention of these results being generalizable to older adults or patient populations. Further, exercise performance was not tested. A knee extension is more of a functional outcome. Please use contractility as that is what was tested. Lastly, please add the limitation that one time point was tested (static measure) or provide data that salivary NO or muscle function is not under the influence of circadian biology. IF it is under the influence of circadian biology than it is necessary that the reader is made aware that any significant percentage change is outside the normal within or between day fluctuation. Given that salivary cortisol does fluctuate that does bring some concern as to how to interpret a static salivary measure.

Reviewer #2: This study investigated the effect of antibacterial mouthwash on muscle contractile function in heathy young adults. The manuscript is well written whilst reviewer has some questions.

1) What is the significance of examining this study in young adults? In previous studies, the effects of NO3- supplementation on muscle function have been observed in elderly people with impaired endothelial function and people with vascular disease.

2) It is questionable whether the methods used in this experiment were appropriate. In this study, no increase in NO activity was obtained with the use of mouthwash. This study is considered to be an inappropriate experimental design, as it would only be valid if the use of mouthwash increases NO activity.

3) How reliable is the measurement of breath NO level for assessing NO activity?

4) Mouthwash was used twice a day, but at what time of the day was it used? Also, was the timing of use standardized for all subjects?

6. PLOS authors have the option to publish the peer review history of their article (what does this mean? ). If published, this will include your full peer review and any attached files.

**Do you want your identity to be public for this peer review?** For information about this choice, including consent withdrawal, please see our Privacy Policy .

Reviewer #1: No

Reviewer #2: No

---

## [Author Response · Author response to Decision Letter 1]

7 Oct 2024

Reviewer #1: General Comments: The authors have provided a well written manuscript focused on investigating the effect of mouthwash on oral biomarkers of nitric oxide bioavailability and lower extremity skeletal muscle function. Although the question is of relevance given the increasing interest in nitrate supplementation, the population for testing their hypothesis is questionable. Given the support provided in the introduction and the exclusionary criteria of several of the author's manuscripts, it would seem that young healthy adults would not benefit from nitrate supplementation, possibly due to the robustness of physiological reserves at that developmental stage. Thus, the blunting of the enterosalivary pathway with mouthwash would need to have large effect to even expect a reduction in skeletal muscle function in this population. As the authors allude in the discussion, studying this effect in older adults or patient populations that have documented poor NO bioavailability would seem to be the ideal population, especially given emerging evidence of links between oral health and frailty. Nonetheless, in accordance with PLOS ONE publication criteria, the study was conducted well, and I only have minor edits about the framing of the results, questionable statistical analyses, and the preliminary nature of the study.

We thank the reviewer for their comments on our manuscript. However, we disagree with their assertion that we tested our hypothesis in the wrong population.

First, although we have published a number of studies of dietary nitrate supplementation in older men and women and patients with heart failure, we have also previously reported that this intervention increases neuromuscular function in healthy young and middle aged untrained men and women (PMCID: PMC4362985) as well as in young athletes (PMCID: PMC4889556). In a follow-up study, we found no apparent effect of age on the magnitude of this effect (PMCID: PMC5789728). This conclusion was supported by our later independent-participant data meta-analysis of the literature (PMCID: PMC8501726), in which the vast majority of participants (i.e., 230 out of 268, or 86%) were young and healthy. It is therefore clear that, contrary to the reviewer’s assertion, dietary nitrate supplementation is beneficial even in individuals such as those included in the present study.

Second, and more importantly, the purpose of the present study was actually to examine the converse question, i.e., to determine whether over-the-counter mouthwash reduced NO levels, and if so, whether this would diminish muscle function. Logically, those most likely to be impacted would be those with normal/adequate levels in the first place – those in whom NO production is already reduced (e.g., older individuals) would presumably be less likely to be impacted (“floor effect”). As it turned out, cetylpyrdinium-based mouthwash seemed to have only a slight influence on salivary markers of NO bioavailability, but this could not have been predicted a priori, as no previous study had tested the effects of cetylpyridinium-containing mouthwash in isolation, only in combination with nitrate supplementation (Ref. 34).

Specific:

Title Page: This should be labelled a pilot randomized trial for several reasons: 1) No sample size calculation was performed and the effect size from the meta-analysis includes older adults, who we would expect to have a large effect; 2) only 13/15 provided saliva, no blood NO biomarkers, and only 8/15 provided breath NO; 3) The one significant result is potentially driven by an outlier (well discuss further in methods).

Per the reviewer’s suggestion, the title of the manuscript has been changed to “Pilot randomized trial of the effect of antibacterial mouthwash on muscle contractile function in healthy young adults”

Abstract: The authors state "Cetylpyridinium mouthwash INHIBITS reduction of nitrate to nitrite...". This was not tested in this study. The mouthwash was associated with a reduction in salivary NO precursors. Please edit accordingly.

The sentence in question has been changed to read “Cetylpyridinium mouthwash reduces the relative abundance of NO2- in the oral cavity . . .”

- Keywords should not duplicate words used in the title. Please revise.

Keywords have been revised.

Methods:

- Inclusion of participants during the COVID 19 pandemic would necessitate record of if they had previously been exposed and potientially controlling for any differences. Long covid impacts are still being tested but some studies suggest initial NO altering effects [PMID: 37657532]. Please add this as a limitation of your study.

Data collection for out study took place in two phases, with n=6 participants studied between November 1st 2019 and March 13th 2020 and n=24 between November 5th 2020 and March 20th 2023. All participants studied during the second phase were required to submit to weekly COVID-19 mitigation testing and then later to be vaccinated as a condition of continued university enrollment. None of the participants experienced a COVID-19 infection during the study, none reported that they were suffering from long COVID, and none reported any reason why they should not perform physical activity (PAR-Q question 7). This information has been added to the manuscript. Nonetheless, we have added the possible impact of COVID-19 as a limitation to the Discussion.

Table 1: Please add Race and Age data. Minorites reports higher exposure to societal stressors that could impact endogenous NO bioavailability. Also, baseline fitness level and body composition were not considered. Given the limitation of BMI and small sample size, differences in aerobic capacity, lean mass, and adipose could impact NO responses. If data is not available, please add as a limitation.

Racial and ethnic characteristics and mean age of the participants have been added to the Methods section. Lack of detailed information on aerobic fitness and body composition have been added to the Discussion as an additional limitation.

Statistical analyses - a sensitivity analysis should be conducted on data provided in figure D. The reader should know if removal of that one individual in the Cetylpyridinium group with the highest pre value changes the statistical significance of the result.

To determine whether our findings were being driven by the results for that single individual, we further analyzed the data two ways:

1) A two-tailed Grubb’s test at P=0.05 was used to test the data for outliers. None were found.

2) The two-way ANOVA was repeated excluding that individual’s results. The decrease in % nitrite in the cetylpyridinium group (i.e., from 22±11% to 17±11%) was still significant (i.e., P=0.0071).

The above information has been added to the manuscript.

Secondly, the effect size cited should be labeled (i.e., is it Cohen's d, partial eta, etc.) and the reader should know why it was selected.

The effect size of 0.15 is actually Cohen’s f statistic, which is the appropriate choice for ANOVA. This is now stated in the manuscript and Cohen’s classic textbook cited. To avoid confusing readers by “mixing and matching” different effect size statistics, we have removed reference to the effect size (Hedge’s g) from our meta-analysis.

Results: The focus on percent change is a bit misleading when the individual data points seem to show that one data point is driving the huge change. Please accompany this with the sensitivity analysis data.

Please see above.

Discussion: Given the limitations the findings should be toned down. Please remove any mention of these results being generalizable to older adults or patient populations. Further, exercise performance was not tested. A knee extension is more of a functional outcome. Please use contractility as that is what was tested. Lastly, please add the limitation that one time point was tested (static measure) or provide data that salivary NO or muscle function is not under the influence of circadian biology. IF it is under the influence of circadian biology than it is necessary that the reader is made aware that any significant percentage change is outside the normal within or between day fluctuation. Given that salivary cortisol does fluctuate that does bring some concern as to how to interpret a static salivary measure.

Per the reviewer’s suggestions, we have removed any reference to the generalizability of our results from the Discussion, and refer to muscle contractility only as a determinant of exercise performance and/or physical function. We have also added references re. circadian changes (or lack thereof) in our outcome measures, and added the fact that we only tested at one time point as a limitation. 

Reviewer #2: This study investigated the effect of antibacterial mouthwash on muscle contractile function in heathy young adults. The manuscript is well written whilst reviewer has some questions.

We thank the reviewer for their compliments re. our writing.

1) What is the significance of examining this study in young adults? In previous studies, the effects of NO3- supplementation on muscle function have been observed in elderly people with impaired endothelial function and people with vascular disease.

Please see response to reviewer 1.

2) It is questionable whether the methods used in this experiment were appropriate. In this study, no increase in NO activity was obtained with the use of mouthwash. This study is considered to be an inappropriate experimental design, as it would only be valid if the use of mouthwash increases NO activity.

The reviewer appears to be confused. As stated in the original manuscript, the purpose of our study was to determine whether over-the-counter mouthwash would reduce (not increase, as twice suggested by the reviewer above) NO bioavailability, and if so, whether this had any impact on muscle contractile function. Obviously inhibiting the enterosalivary pathway of NO production via mouthwash use would not be expected to increase NO levels.

3) How reliable is the measurement of breath NO level for assessing NO activity?

Breath NO levels are a well-established and sensitive indicator of changes in whole-body NO production. A reference has been added to this effect.

4) Mouthwash was used twice a day, but at what time of the day was it used? Also, was the timing of use standardized for all subjects?

Participants were instructed to follow the mouthwash manufacturers’ recommendations, which were to use the product morning and evening. This detail has been added to the revised manuscript. Aside from verifying AM and PM use, we did not record the exact timing. As originally stated, however, the effects of cetylpyridinium persist for at least 6 h, which would seem to make the exact timing relatively unimportant.

---

## [Decision Letter · Decision Letter 1]

1 Nov 2024

PONE-D-24-32002R1Pilot randomized trial of the effect of antibacterial mouthwash on muscle contractile function in healthy young adultsPLOS ONE

Dear Dr. Coggan,

Thank you for submitting your manuscript to PLOS ONE. After careful consideration, we feel that it has merit but does not fully meet PLOS ONE’s publication criteria as it currently stands. Therefore, we invite you to submit a revised version of the manuscript that addresses the points raised during the review process.

We look forward to receiving your revised manuscript.

Kind regards,

Ajaya Bhattarai

Academic Editor

PLOS ONE

Journal Requirements:

**Additional Editor Comments:**

Revision is needed on the revised manuscript.

Reviewers' comments:

Reviewer's Responses to Questions

**Comments to the Author**

1. If the authors have adequately addressed your comments raised in a previous round of review and you feel that this manuscript is now acceptable for publication, you may indicate that here to bypass the “Comments to the Author” section, enter your conflict of interest statement in the “Confidential to Editor” section, and submit your "Accept" recommendation.

Reviewer #1: All comments have been addressed

Reviewer #2: All comments have been addressed

Reviewer #3: (No Response)

2. Is the manuscript technically sound, and do the data support the conclusions?

Reviewer #1: Yes

Reviewer #2: Partly

Reviewer #3: Partly

3. Has the statistical analysis been performed appropriately and rigorously? 

Reviewer #1: Yes

Reviewer #2: Yes

Reviewer #3: No

4. Have the authors made all data underlying the findings in their manuscript fully available?

Reviewer #1: Yes

Reviewer #2: Yes

Reviewer #3: Yes

5. Is the manuscript presented in an intelligible fashion and written in standard English?

Reviewer #1: Yes

Reviewer #2: Yes

Reviewer #3: Yes

6. Review Comments to the Author

Reviewer #1: This reviewer is appreciative of the author's responses and believe this manuscript less confusing in its current state. There are only 2 new issues with the manuscript that should be address prior to publication. Well done.

1) As it currently stands, although the overall values are representative of the local population, it is unclear how the 4 minorities are dispersed across the experimental groups. Please add racial/ethnic data and group sample size to table 1 and perform a statistical group comparison and add appropriate p-values to table 1.

2) This study does not provide conclusive data to make this claim in line the final paragraph of the manuscript: "Healthy young Athletes seemingly need not be concerned about possible impairments in muscle contractility due to use

of cetylpyridinium-based mouthwash.". Please remove or add terms of uncertainty.

Reviewer #2: Revised manuscript and the responses to comments were reviewed. I found them to be generally satisfactory.

Reviewer #3: This is a quite an interesting study. However, the manuscript could benefit from further improvements based on the following comments:

Page 4: Apart from the exclusion criteria, inclusion criteria is to be provided.

Page 4: The study design name is to be clearly stated.

Page 5 Paragraph 1 Line 4: The name abbreviation could be attached to the 'individual' as a form of identification.

Pahe 5: The baseline, intervention and post time period is to be clearly stated.

Page 5 Paragraph 1 Line 2: The cut off/requirement or level of acceptance PAR-Q and isokinetic dynamometer is to be stated. The university name is to be stated.

Page 5 Table 1: The table presentation to follow journal format. The decimal point is to be standardized. Title is too short.

Page 7: ‘the latter after log normalization’ is to be rephrased.

Page 7 Paragraph 2 Line 4 & 6: two-way analysis of variance (ANOVA) and three-way analysis of variance (ANOVA) are to be revised to two-way repeated measure analysis of variance (ANOVA) and three-way repeated measure analysis of variance (ANOVA) respectively. The detail analysis in table form is to be presented e.g. main effects, interactions, effect size, post hoc comparison etc and attached as supplementary table. The statistical test assumptions fulfilment is to be stated.

Page 7, Paragraph 2, Line 7: The sentence ‘with treatment as a between-subject factor and velocity and time as within-subject factors’ can be revised to ‘with treatment as a between-subject factor while velocity and time are treated as within-subject factors.

Page 7 Paragraph 2 Line 8: One or two-tailed test for p value is to be stated.

Page 7: The statement ‘According to Cohen [32], this would reflect a “moderate” effect size’ unclear and requires further elaboration. Based on Cohen’s guidelines, Cohen f = 0.15 is considered small and 0.25 considered moderate effect size. 0.15 -0.25 considered small to moderate effect size.

Page 7, Paragraph 2: Missing data (if any) is to be reported. If there is no missing data, a statement indicating this is to be included.

Page 8 Paragraph 2 Line 5: Indicate clearly the p value in Figure 2D.

Page 8 Paragraph 3 Line 1-3 Muscle contractile function: The results (e.g. p values) are to be clearly presented in the table. For the statement ‘nor were there any significant interaction effects (Table 2)’, the results are to be presented.

Table 2: for the table footnote ‘No significant treatment effects were observed’, the p value (s) is/are to be stated/presented. Statistical test is to be denoted in the table footnote. More detailed information such as mean difference, effect size index, p values, 95%CI etc could be presented.

7. PLOS authors have the option to publish the peer review history of their article (what does this mean? ). If published, this will include your full peer review and any attached files.

**Do you want your identity to be public for this peer review?** For information about this choice, including consent withdrawal, please see our Privacy Policy .

Reviewer #1: No

Reviewer #2: No

Reviewer #3: No

---

## [Author Response · Author response to Decision Letter 2]

15 Nov 2024

Reviewer #1: This reviewer is appreciative of the author's responses and believe this manuscript less confusing in its current state. There are only 2 new issues with the manuscript that should be address prior to publication. Well done.

1) As it currently stands, although the overall values are representative of the local population, it is unclear how the 4 minorities are dispersed across the experimental groups. Please add racial/ethnic data and group sample size to table 1 and perform a statistical group comparison and add appropriate p-values to table 1.

Table 1 has been expanded to provide the information and statistical comparisons requested.

2) This study does not provide conclusive data to make this claim in line the final paragraph of the manuscript: "Healthy young Athletes seemingly need not be concerned about possible impairments in muscle contractility due to use of cetylpyridinium-based mouthwash.". Please remove or add terms of uncertainty.

The offending sentence has been replaced with “Short-term use of cetylpyridinium-based mouthwash does not seem to impair whole-body NO bioavailability or muscle contractility in healthy young Individuals.” Note that this is essentially just a restatement of the results.

Reviewer #3: This is a quite an interesting study. However, the manuscript could benefit from further improvements based on the following comments:

Page 4: Apart from the exclusion criteria, inclusion criteria is to be provided.

As succinctly stated in the original manuscript, “All other persons were included in the study.” We believe that this already makes it clear that anyone between 18 and 30 y of age, who was not currently using mouthwash, antibiotics, or tobacco products, who had a resting blood pressure > 140/90, who answered no to all of the questions of the Physical Activity Readiness Questionnaire (PAR-Q), and was able to provide informed consent was eligible for inclusion. No changes have therefore been made.

Page 4: The study design name is to be clearly stated.

That a parallel-arm design was used is now explicitly stated in the last sentence of the Introduction.

Page 5 Paragraph 1 Line 4: The name abbreviation could be attached to the 'individual' as a form of identification.

The initials of the individual responsible for generating the treatment assignment have been added.

Pahe 5: The baseline, intervention and post time period is to be clearly stated.

It is unclear what the reviewer expects, as this was an acute study, with the intervention period (i.e., 7 d) already clearly stated. Nonetheless, in response we have added the word “immediately” to indicate that there was no delay between the baseline testing and treatment period, or between the treatment period and post-testing.

Page 5 Paragraph 1 Line 2: The cut off/requirement or level of acceptance PAR-Q and isokinetic dynamometer is to be stated. The university name is to be stated.

The name of the university was already explicitly stated just two sentences above, but in response to the reviewer’s comment has also been added here. However, it is completely unclear what they mean by “the cut off/requirement or level of acceptance PAR-Q and isokinetic dynamometer “, so no other changes have been made.

Page 5 Table 1: The table presentation to follow journal format. The decimal point is to be standardized. Title is too short.

We have read carefully the guidelines for table formatting (https://journals.plos.org/plosone/s/tables) and believe that our tables are in conformance. No changes have therefore been made.

Page 7: ‘the latter after log normalization’ is to be rephrased.

This sentence has been revised to read “Salivary NO3-, NO2-, and log-normalized cGMP concentrations . . .”

Page 7 Paragraph 2 Line 4 & 6: two-way analysis of variance (ANOVA) and three-way analysis of variance (ANOVA) are to be revised to two-way repeated measure analysis of variance (ANOVA) and three-way repeated measure analysis of variance (ANOVA) respectively. The detail analysis in table form is to be presented e.g. main effects, interactions, effect size, post hoc comparison etc and attached as supplementary table. The statistical test assumptions fulfilment is to be stated.

Please see below.

Page 7, Paragraph 2, Line 7: The sentence ‘with treatment as a between-subject factor and velocity and time as within-subject factors’ can be revised to ‘with treatment as a between-subject factor while velocity and time are treated as within-subject factors.

Both of these comments actually refer to the same two sentences describing the statistical analysis of the data. We believe the fact that repeated measures ANOVAs were performed was already clearly apparent from the nature of the study design and the fact that velocity and time were treated as within-subject factors. Nonetheless, in response to the reviewer’s comment we have now added the words “repeated measures” to both sentences. Full results of these analyses are also now included as a supplement.

Page 7 Paragraph 2 Line 8: One or two-tailed test for p value is to be stated.

The use of two-tailed tests is now specified.

Page 7: The statement ‘According to Cohen [32], this would reflect a “moderate” effect size’ unclear and requires further elaboration. Based on Cohen’s guidelines, Cohen f = 0.15 is considered small and 0.25 considered moderate effect size. 0.15 -0.25 considered small to moderate effect size.

According to Cohen (pages 285-286 of Ref. 32), f = 0.10 (not 0.15) is considered a “small” effect size. The sentence in question has therefore been revised to read “According to Cohen [32], this would be between a “small” (f = 0.10) and a “medium” (f = 0.25) effect size.”

Page 7, Paragraph 2: Missing data (if any) is to be reported. If there is no missing data, a statement indicating this is to be included.

This has been addressed in the following paragraph.

Page 8 Paragraph 2 Line 5: Indicate clearly the p value in Figure 2D.

The P value is clearly shown at the top right of panel D, as well as stated in the text.

Page 8 Paragraph 3 Line 1-3 Muscle contractile function: The results (e.g. p values) are to be clearly presented in the table. For the statement ‘nor were there any significant interaction effects (Table 2)’, the results are to be presented.

Please see below.

Table 2: for the table footnote ‘No significant treatment effects were observed’, the p value (s) is/are to be stated/presented. Statistical test is to be denoted in the table footnote. More detailed information such as mean difference, effect size index, p values, 95%CI etc could be presented.

In response to the reviewer’s comments, P values for isometric/isokinetic torque have been moved from the text to the table, and interaction as well as main effects provided. These changes have made the footnote redundant, so it has been deleted. Conversely, main and interactions effects for Vmax and Pmax have been added to the text for completeness.

---

## [Decision Letter · Decision Letter 2]

8 Dec 2024

PONE-D-24-32002R2Pilot randomized trial of the effect of antibacterial mouthwash on muscle contractile function in healthy young adultsPLOS ONE

Dear Dr. Coggan,

Thank you for submitting your manuscript to PLOS ONE. After careful consideration, we feel that it has merit but does not fully meet PLOS ONE’s publication criteria as it currently stands. Therefore, we invite you to submit a revised version of the manuscript that addresses the points raised during the review process.

We look forward to receiving your revised manuscript.

Kind regards,

Ajaya Bhattarai

Academic Editor

PLOS ONE

Journal Requirements:

Additional Editor Comments:

Minor revision is needed.

Reviewers' comments:

Reviewer's Responses to Questions

**Comments to the Author**

1. If the authors have adequately addressed your comments raised in a previous round of review and you feel that this manuscript is now acceptable for publication, you may indicate that here to bypass the “Comments to the Author” section, enter your conflict of interest statement in the “Confidential to Editor” section, and submit your "Accept" recommendation.

Reviewer #3: All comments have been addressed

Reviewer #4: (No Response)

2. Is the manuscript technically sound, and do the data support the conclusions?

Reviewer #3: Partly

Reviewer #4: Yes

3. Has the statistical analysis been performed appropriately and rigorously? 

Reviewer #3: Yes

Reviewer #4: Yes

4. Have the authors made all data underlying the findings in their manuscript fully available?

Reviewer #3: Yes

Reviewer #4: Yes

5. Is the manuscript presented in an intelligible fashion and written in standard English?

Reviewer #3: Yes

Reviewer #4: Yes

6. Review Comments to the Author

Reviewer #3: Minor comment

For Table 1, based on CONSORT statement, statistical tests to compare baseline characteristics between intervention groups are not recommended. Instead, the CONSORT guidelines recommend presenting the baseline data descriptively. As such statistical tests/p values are to be omitted.

Reviewer #4: The authors in the study examined the effect of antibacterial mouthwash on muscle contractile function in healthy young adults. This study is well done and the manuscript has been revised based on reviewer comments.

I have only two minor comments

P.6. An examination of saliva samples. Typically, during ELISA analysis, duplicate sample analysis is employed for data evaluation. It is essential to incorporate these data regarding the coefficient of variations.

Results

You should add about F value in results of analysis of ANOVA.

7. PLOS authors have the option to publish the peer review history of their article (what does this mean? ). If published, this will include your full peer review and any attached files.

**Do you want your identity to be public for this peer review?** For information about this choice, including consent withdrawal, please see our Privacy Policy .

Reviewer #3: No

Reviewer #4: No

---

## [Author Response · Author response to Decision Letter 3]

7 Jan 2025

Reviewer #3: Minor comment

For Table 1, based on CONSORT statement, statistical tests to compare baseline characteristics between intervention groups are not recommended. Instead, the CONSORT guidelines recommend presenting the baseline data descriptively. As such statistical tests/p values are to be omitted.

Statistical comparisons between groups have been removed.

Reviewer #4: The authors in the study examined the effect of antibacterial mouthwash on muscle contractile function in healthy young adults. This study is well done and the manuscript has been revised based on reviewer comments.

I have only two minor comments

P.6. An examination of saliva samples. Typically, during ELISA analysis, duplicate sample analysis is employed for data evaluation. It is essential to incorporate these data regarding the coefficient of variations.

The CV for these triplicate determinations has now been stated.

Results

You should add about F value in results of analysis of ANOVA.

Added.

---

## [Decision Letter · Decision Letter 3]

14 Jan 2025

PONE-D-24-32002R3Pilot randomized trial of the effect of antibacterial mouthwash on muscle contractile function in healthy young adultsPLOS ONE

Dear Dr. Coggan,

Thank you for submitting your manuscript to PLOS ONE. After careful consideration, we feel that it has merit but does not fully meet PLOS ONE’s publication criteria as it currently stands. Therefore, we invite you to submit a revised version of the manuscript that addresses the points raised during the review process.

We look forward to receiving your revised manuscript.

Kind regards,

Ajaya Bhattarai

Academic Editor

PLOS ONE

Journal Requirements:

Reviewers' comments:

Reviewer's Responses to Questions

**Comments to the Author**

1. If the authors have adequately addressed your comments raised in a previous round of review and you feel that this manuscript is now acceptable for publication, you may indicate that here to bypass the “Comments to the Author” section, enter your conflict of interest statement in the “Confidential to Editor” section, and submit your "Accept" recommendation.

Reviewer #3: All comments have been addressed

Reviewer #4: All comments have been addressed

2. Is the manuscript technically sound, and do the data support the conclusions?

Reviewer #3: Partly

Reviewer #4: Yes

3. Has the statistical analysis been performed appropriately and rigorously? 

Reviewer #3: Yes

Reviewer #4: Yes

4. Have the authors made all data underlying the findings in their manuscript fully available?

Reviewer #3: Yes

Reviewer #4: Yes

5. Is the manuscript presented in an intelligible fashion and written in standard English?

Reviewer #3: Yes

Reviewer #4: Yes

6. Review Comments to the Author

Reviewer #3: Minor comment

Table 1 footnote: The statement 'P values calculated using Fisher’s exact test and two-tailed unpaired t tests, as

appropriate.' is to be omitted.

Reviewer #4: (No Response)

7. PLOS authors have the option to publish the peer review history of their article (what does this mean? ). If published, this will include your full peer review and any attached files.

**Do you want your identity to be public for this peer review?** For information about this choice, including consent withdrawal, please see our Privacy Policy .

Reviewer #3: No

Reviewer #4: No

---

## [Author Response · Author response to Decision Letter 4]

14 Jan 2025

Reviewer #3: Minor comment

Table 1 footnote: The statement 'P values calculated using Fisher’s exact test and two-tailed unpaired t tests, as appropriate.' is to be omitted.

The leftover footnote has been removed. (Why the reviewer insisted on further holding acceptance/publication of the paper versus simply letting this be handled at the copyediting stage is completely beyond me.)

---

## [Decision Letter · Decision Letter 4]

16 Jan 2025

Pilot randomized trial of the effect of antibacterial mouthwash on muscle contractile function in healthy young adults

PONE-D-24-32002R4

Dear Dr. Coggan,

We’re pleased to inform you that your manuscript has been judged scientifically suitable for publication and will be formally accepted for publication once it meets all outstanding technical requirements.

Kind regards,

Ajaya Bhattarai

Academic Editor

PLOS ONE

Additional Editor Comments (optional):

Reviewers' comments:

Reviewer's Responses to Questions

**Comments to the Author**

1. If the authors have adequately addressed your comments raised in a previous round of review and you feel that this manuscript is now acceptable for publication, you may indicate that here to bypass the “Comments to the Author” section, enter your conflict of interest statement in the “Confidential to Editor” section, and submit your "Accept" recommendation.

Reviewer #3: All comments have been addressed

2. Is the manuscript technically sound, and do the data support the conclusions?

Reviewer #3: Partly

3. Has the statistical analysis been performed appropriately and rigorously? 

Reviewer #3: Yes

4. Have the authors made all data underlying the findings in their manuscript fully available?

Reviewer #3: Yes

5. Is the manuscript presented in an intelligible fashion and written in standard English?

Reviewer #3: Yes

6. Review Comments to the Author

Reviewer #3: (No Response)

7. PLOS authors have the option to publish the peer review history of their article (what does this mean? ). If published, this will include your full peer review and any attached files.

**Do you want your identity to be public for this peer review?** For information about this choice, including consent withdrawal, please see our Privacy Policy .

Reviewer #3: No

---

## [Editor Report · Acceptance letter]

PONE-D-24-32002R4

PLOS ONE

Dear Dr. Coggan,

I'm pleased to inform you that your manuscript has been deemed suitable for publication in PLOS ONE. Congratulations! Your manuscript is now being handed over to our production team.

Kind regards,

on behalf of

Dr. Ajaya Bhattarai

Academic Editor

PLOS ONE